# Simulating Future LUCC by Coupling Climate Change and Human Effects Based on Multi-Phase Remote Sensing Data

**Zihao Huang** [1,2,3], **Xuejian Li** [1,2,3], **Huaqiang Du** [1,2,3,*], **Fangjie Mao** [1,2,3], **Ning Han** [1,2,3], **Weiliang Fan** [1,2,3], **Yanxin Xu** [1,2,3] and **Xin Luo** [1,2,3]

1   State Key Laboratory of Subtropical Silviculture, Zhejiang A & F University, Hangzhou 311300, China; huangzihao@stu.zafu.edu.cn (Z.H.); xuejianli201609@163.com (X.L.); mfangjie@gmail.com (F.M.); hangis2002@163.com (N.H.); fanweiliang@zafu.edu.cn (W.F.); xuyanxin@stu.zafu.edu.cn (Y.X.); luoxin@stu.zafu.edu.cn (X.L.)
2   Key Laboratory of Carbon Cycling in Forest Ecosystems and Carbon Sequestration of Zhejiang Province, Zhejiang A & F University, Hangzhou 311300, China
3   School of Environmental and Resources Science, Zhejiang A & F University, Hangzhou 311300, China
*   Correspondence: dahuaqiang@zafu.edu.cn

**Abstract:** Future land use and cover change (LUCC) simulations play an important role in providing fundamental data to reveal the carbon cycle response of forest ecosystems to LUCC. Subtropical forests have great potential for carbon sequestration, yet their future dynamics under natural and human influences are unclear. Zhejiang Province in China is an important distribution area for subtropical forests. For forest management, it is of great significance to explore the future dynamic changes of subtropical forests in Zhejiang. As a popular LUCC spatial simulation model, the cellular automata (CA) model coupled with machine learning and LUCC quantitative demand models such as system dynamics (SD) can achieve effective LUCC simulation. Therefore, we first integrated a back propagation neural network (BPNN), a CA, and a SD model as a BPNN_CA_SD (BCS) coupled model for future LUCC simulation and then designed a slow development scenario (SD_Scenario), a harmonious development scenario (HD_Scenario), a baseline development scenario (BD_Scenario), and a fast development scenario (FD_Scenario), combining climate change and human disturbance. Thirdly, we obtained future land-use patterns in Zhejiang Province from 2014 to 2084 under multiple scenarios, and finally, we analyzed the temporal and spatial changes of land use and discussed the subtropical forest dynamics of the future. The results showed the following: (1) The overall accuracy was approximately 0.8, the kappa coefficient was 0.75, and the figure of merit (FOM) value was over 28% when using the BCS model to predict LUCC, indicating that the model could predict the consistent change of LUCC accurately. (2) The future evolution of the LUCC under different scenarios varied, with the growth of bamboo forests and the decline of coniferous forests in the FD_Scenario being prominent among the forest dynamics changes. Compared with 2014, the bamboo forest in 2084 will increase by 37%, while the coniferous forest will decrease by 25%. (3) Comparing the area and spatial change of the subtropical forests, the SD_Scenario was found to be beneficial for the forest ecology. These results can provide an important decision-making reference for land-use planning and sustainable forest development in Zhejiang Province.

**Keywords:** land use and land cover change (LUCC); scenario simulation; cellular automata (CA) model; system dynamics (SD) model; back propagation neural network (BPNN)

## 1. Introduction

Land use and land cover change (LUCC) is a direct driving factor of the carbon balance in terrestrial ecosystems, and its impact on global warming is second only to that of fossil fuels and industrial emissions [1–4]. Therefore, the impact of LUCC and climate change on forest spatiotemporal dynamics has been widely appreciated. However, limited LUCC data may lead to significant underestimations within the impact of LUCC on carbon

emissions [5], and the absence of future LUCC data under future climate change is a major limitation for revealing the response of forest ecosystems' carbon cycles to future climate change [6]. Therefore, it is of great scientific significance to obtain future LUCC data through spatiotemporal LUCC simulations and to explore the LUCC evolution in order to reveal the impact of LUCC on the forest ecosystem's carbon cycle.

There are the following three types of models to simulate LUCC: quantitative models, spatial models, and coupled models of the first two. Quantitative demand models focus on predicting the area transfer among different land-use types, including system dynamics (SD), grey prediction, and Markov chain models [7–9]. Most of these models are statistical models. The SD model is a simulation method in which inventories and flows, with corresponding feedback loops, are used to simulate large-scale, complex socioeconomic systems [10]. SD models seek to understand how physical processes, information flows, and management policies interact [11] so as to link feedback between different variables in the system and, therefore, it can be used to deal with dynamic processes [12]. Previous studies have been conducted to show that the SD model for LUCC demand prediction is the most effective of the quantitative models, mainly because it considers comprehensive factors such as the impact of markets, policies, and climate adaptation strategies [13]. However, these quantitative models lack the ability to project the spatial patterns of land use; thus, spatial models that reveal LUCC at a spatial scale are required. The cellular automata (CA) model and the conversion of land use and its effects at a small regional extent (CLUE-S) model are popular methods to simulate the LUCC spatial evolution [12,14–17]. These models with spatial function are all scale-dependent. Among these models, the CA model has simple preconditions and rules expressed by a matrix where a state defines each cell [10,15]. It can simulate spatial dynamics from a bottom-up perspective and is capable of establishing interconnections between LUCC and driving forces [18,19]. Its main advantages are in expressing the driving forces of LUCC by transition rules and expressing spatial externalities by neighborhood effects [20,21]. In addition, it is easily combined with regression models to identify locations with high suitability for LUCC based on weighted overlay suitability factors [17,22,23]. This has become an excellent method for predicting future LUCC according to different scenarios [24,25].

However, the single model described above could not fully consider the internal mechanisms of the ecosystem when simulating LUCC. Therefore, the coupled model integrated by a top-down quantitative model and a bottom-up spatial model has been popular to improve the accuracy of LUCC simulation [7]. The FLUS model could be applied to the effective Chinese LUCC simulation considering various socioeconomic and natural climatic factors, which are integrated by an artificial neural network, a CA model, and an SD model [26]. Considering that the performance of the back propagation neural network (BPNN) is better than some artificial neural networks [27], the CA model integrated with the BPNN (BPNN_CA) can be designed to control the spatial pattern changes more precisely. Furthermore, the interactive coupling of the CA model and the SD model is more effective and popular than the loose coupling of the two. Therefore, it is achievable to interactively couple the three models of BPNN, CA, and SD together as a BPNN_CA_SD (BCS) model for simulating LUCC precisely. Gradually, the refinement of coupled models meets the accuracy requirements of LUCC spatiotemporal simulation.

Subtropical forests, with their diverse types, high photosynthetic capacity, and four-season growth, have been of global concern for their carbon sequestration capacity, which accounts for approximately 40% of the world's gross primary productivity together with tropical forests [28,29]. China is an important distribution area for subtropical forests. The vegetation carbon storage of the evergreen broad-leaved forest and coniferous and broad-leaved mixed forest in subtropical China is 2.527 billion tons, accounting for approximately 30% of the country [30]. The carbon sink here has great potential, but it is very sensitive to LUCC caused by global climate change and anthropogenic disturbances. Regarding the impact of LUCC on the subtropical carbon cycle, its past and current impacts have been studied [31,32], while the impact of LUCC is still unknown under future

scenarios [33,34]. Simultaneously, future spatiotemporal LUCC modeling studies mainly focus on urban expansion [35,36], while less research has been conducted on the future distribution of subtropical forests, and the complex competitive relationships between subtropical forests and other land-use types are not yet clear. Given this situation, we posed our research questions as follows: (1) How to establish a coupled model that takes into account natural and human influences that can accurately simulate the quantitative and spatial changes of subtropical forests? (2) How will subtropical forests change in the future under natural and human effects, and which scenarios are beneficial to subtropical forests?

As a typical region of subtropical forests, Zhejiang Province has one of the highest forest coverage rates in China, and its forest changes have been of great concern. Simultaneously, the Coupled Model Intercomparison Project 5 (CMIP5) stage could provide reasonable scenario data for projecting future LUCC dynamics by combining representative concentration pathways (RCPs) 2.6, 4.5, 6.0, and 8.5 scenarios [37,38]. RCPs are used for scenario modeling under natural influences such as greenhouse gas emissions and radiative forcing. The combination of RCPs and socioeconomic development scenarios into integrated scenarios will make future LUCC projections more reasonable and accurate [39]. Thus, in this study, we take Zhejiang Province as an example to develop a BPNN_CA_SD (BCS) coupled model for future LUCC simulation under multiple development scenarios. The objectives consisted of the following four specific parts: (1) to build an SD model to project the land-use area demands, (2) to establish a BPNN_CA model to deal with the simulation of the land-use spatial patterns, (3) to interactively couple BPNN_CA with SD as the BCS model to obtain future spatiotemporal land-use patterns under the multiple natural and socioeconomic development scenarios we set, and (4) to analyze the LUCC of Zhejiang Province under different scenarios and provide a scenario reference for managing subtropical forests in response to the impacts of climate change and human disturbances.

## 2. Materials and Methods

### 2.1. Study Area

Zhejiang Province (Figure 1a) is located in the south of Yangtze River Delta on the southeast coast of China, with a total area of $1.055 \times 10^5$ km$^2$. Zhejiang Province includes 11 administrative cities (Figure 1b), including Hangzhou, Ningbo, and Wenzhou. The province belongs to a subtropical monsoon climate. It has rich forest resources, and the forest coverage rate of the whole province has been increasing since 2004 and reached 61.15% in 2020 (Figure 1c). The major forest types are broad-leaved forest, coniferous forest, and bamboo forest (Figure 1d).

### 2.2. Data Sets and Processing

Natural environmental factors and socioeconomic factors have been proposed as the two main driving forces of land use/cover change by the IGBP and IHDP [40]. To simulate how climate change and human activities affect LUCC, these two types of driving forces (Table 1) were considered and consisted of the following three components: (1) geospatial data were used for spatial simulation. All the geospatial data were projected onto a WGS_1984 coordinate system and resampled to the same resolution ($30 \times 30$ m) of land-use patterns. (2) Macro statistical data were used for quantitative simulation. (3) Sample plotted data were the classification verification data of each land-use type, which were used to verify the accuracy.

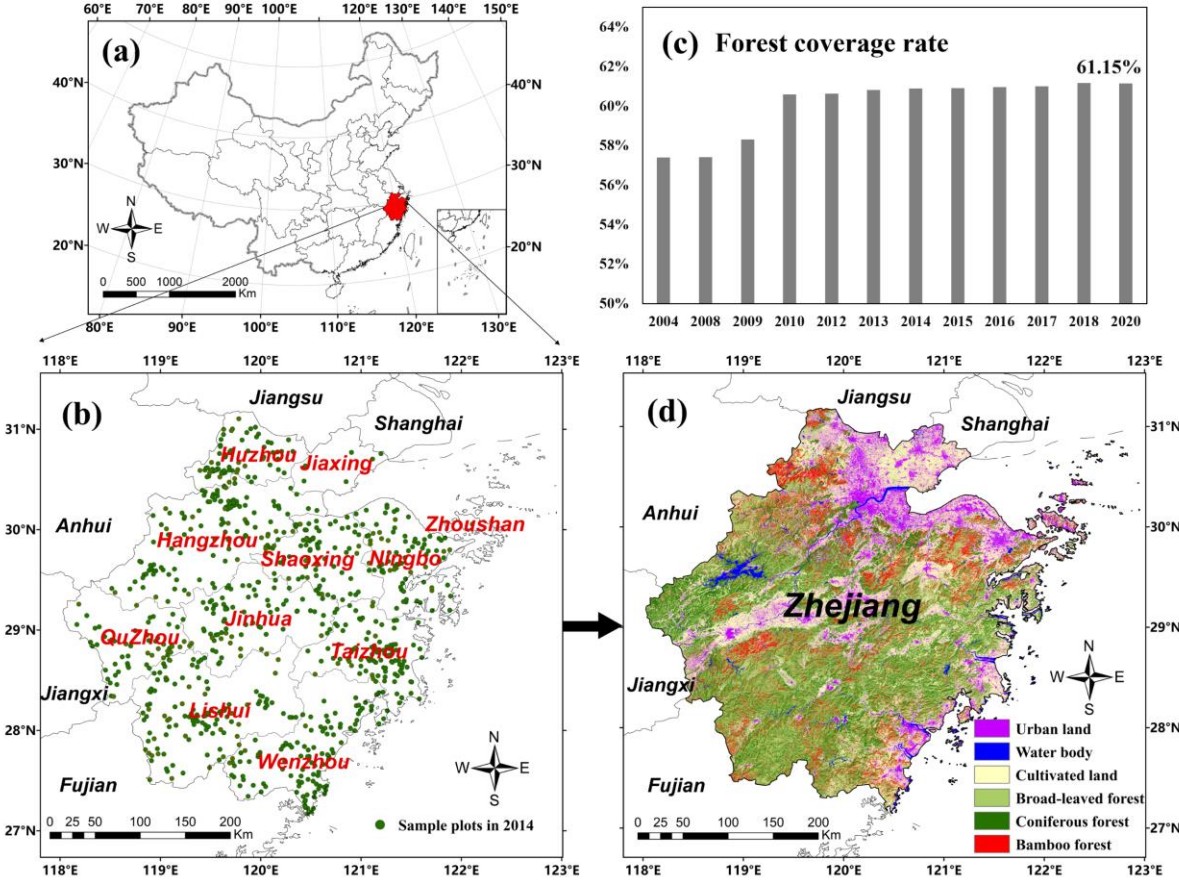

**Figure 1.** Zhejiang Province study area: (**a**) location in China, (**b**) sample plots in 2014, (**c**) forest coverage rate from 2004 to 2020, and (**d**) land-use patterns in 2014.

**Table 1.** List of data used in this study.

| Data Set | Category | Data | Year | Resolution | Diagram | Data Resource |
|---|---|---|---|---|---|---|
| Geo-spatial data | Land use | Land-use patterns | 1984–2014 | 30 m | Figure 2a,b | The data were based on the Satellite 30 m multispectral data of Landsat-5 TM (1984–2008) and Landsat-8 OLI (2014). After radiation correction, atmospheric correction, and geometric correction, the maximum likelihood classification method was used to obtain land-use patterns. |
| | Terrain | DEM | 2014 | 30 m | Figure 3a–c | Downloaded from the Geospatial Data Cloud site (http://www.gscloud.cn, accessed on 24 December 2021). |
| | | Slope | | | | Calculated from DEM. |
| | | Aspect | | | | |
| | Soil | Silt fraction | 2008 | 1 km | Figure 3d–i | Silt fraction, clay fraction, sand fraction, and available water content were derived from the Harmonized World Soil Database (HWSD 1.2). The bulk density and soil wilt point were calculated by the silt and clay fraction [41]. |
| | | Clay fraction | | | | |
| | | Sand fraction | | | | |
| | | Available water content | | | | |
| | | Bulk density | | | | |
| | | Wilt point | | | | |
| | Climate | Total precipitation | 1984–2014 | 1 km | Figure 3j–m | The annual data were calculated from the averages or sums of the daily data. The daily data were interpolated from observations at 410 meteorological stations in Zhejiang Province and its surrounding provinces using the inverse distance weighted method [42]. |
| | | Average temperature | | | | |
| | | Average radiation | | | | |
| | | Average relative humidity | | | | |
| | Human influence | Population | 2015 | 1 km | Figure 3n,o | Obtained from the Resource and Environmental Science and Data Center of the Chinese Academy of Sciences (http://www.resdc.cn, accessed on 24 December 2021). |
| | | Gross domestic product (GDP) | | | | |
| | | Distance to roads | 2014 | 30 m | Figure 3p–r | Calculated from the vector maps of the roads, the railways, and the water systems, which were downloaded from the Open Street map (https://www.openstreetmap.org/, accessed on 12 October 2020). |
| | | Distance to railways | | | | |
| | | Distance to water | | | | |

**Table 1.** *Cont.*

| Data Set | Category | Data | Year | Resolution | Diagram | Data Resource |
|---|---|---|---|---|---|---|
| Macro statistics data | | Land-use area | 1984–2014 | - | - | Calculated from the land-use patterns. |
| | | Total precipitation statistics | | | - | Calculated from the total precipitation. |
| | | Average temperature statistics | | | - | Calculated from the average temperature. |
| | | Population statistics | | | Figure 4a–d | Collected from the Zhejiang Statistical Yearbook (http://tjj.zj.gov.cn/, accessed on 12 October 2020). |
| | | GDP statistics | | | | |
| | | Grain yield | | | | |
| | | Aquatic product yield | | | | |
| | | Forest coverage rate | 2004–2020 | | Figure 1c | Collected from the Announcement of Forest Resources and Its Ecological Function Value of Zhejiang Province (http://lyj.zj.gov.cn/index.html, accessed on 24 December 2021). |
| Sample plots data | | Classification verification plots | 1984–2014 | - | Figure 1b and Table 2 | Classification verification plots of BLF, CF, and BF were derived from the data of the National Forest Inventory. Verification plots of other land-use types were based on field investigation and image visual interpretation. |

The land use of Zhejiang Province (Figure 2a) was classified into the following six types: urban land (UL), water body (WB), cultivated land (CL), broad-leaved forest (BLF), coniferous forest (CF), and bamboo forest (BF) [43,44]. The sample plots data (Tables 1 and 2) were used to calculate the overall accuracy (OA) and kappa coefficient (Kappa) to evaluate the accuracy of the classification results. Figure 2b shows the normalized confusion matrices from 1984 to 2014. Each value of the main diagonal in the normalized confusion matrix corresponds to the producer's accuracy (PA) value for each land-use type. It can be seen that the overall accuracy (OA) and kappa coefficients (Kappa) in different years were all higher than 0.78 and 0.73, respectively, indicating that the classification results were highly consistent with the actual situation. Moreover, most of the PA values were higher than 76%, which provides an important guarantee for spatiotemporal LUCC simulation in the study area.

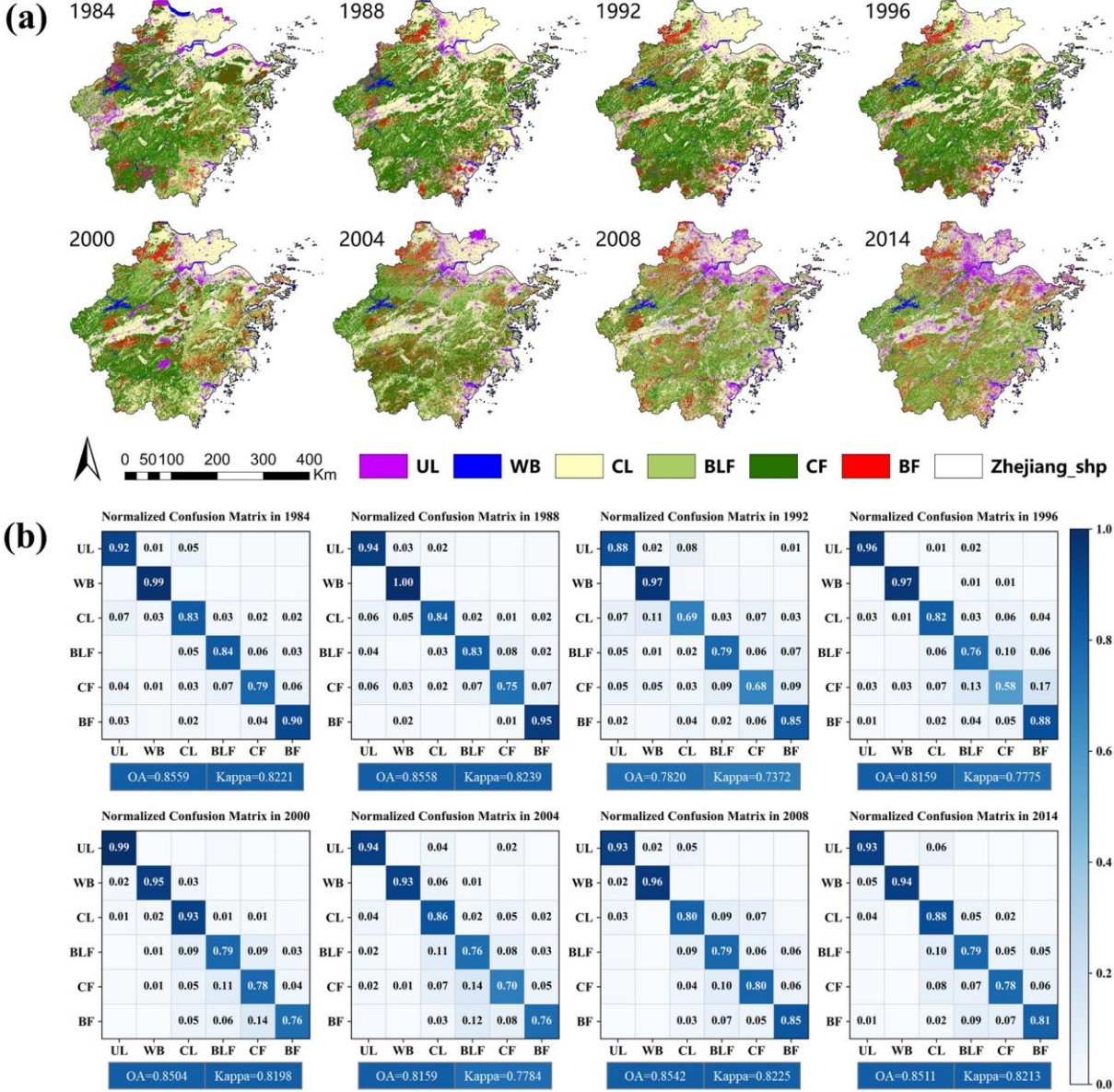

**Figure 2.** (**a**) Land-use patterns and (**b**) accuracy evaluation of land-use classification in Zhejiang Province from 1984 to 2014.

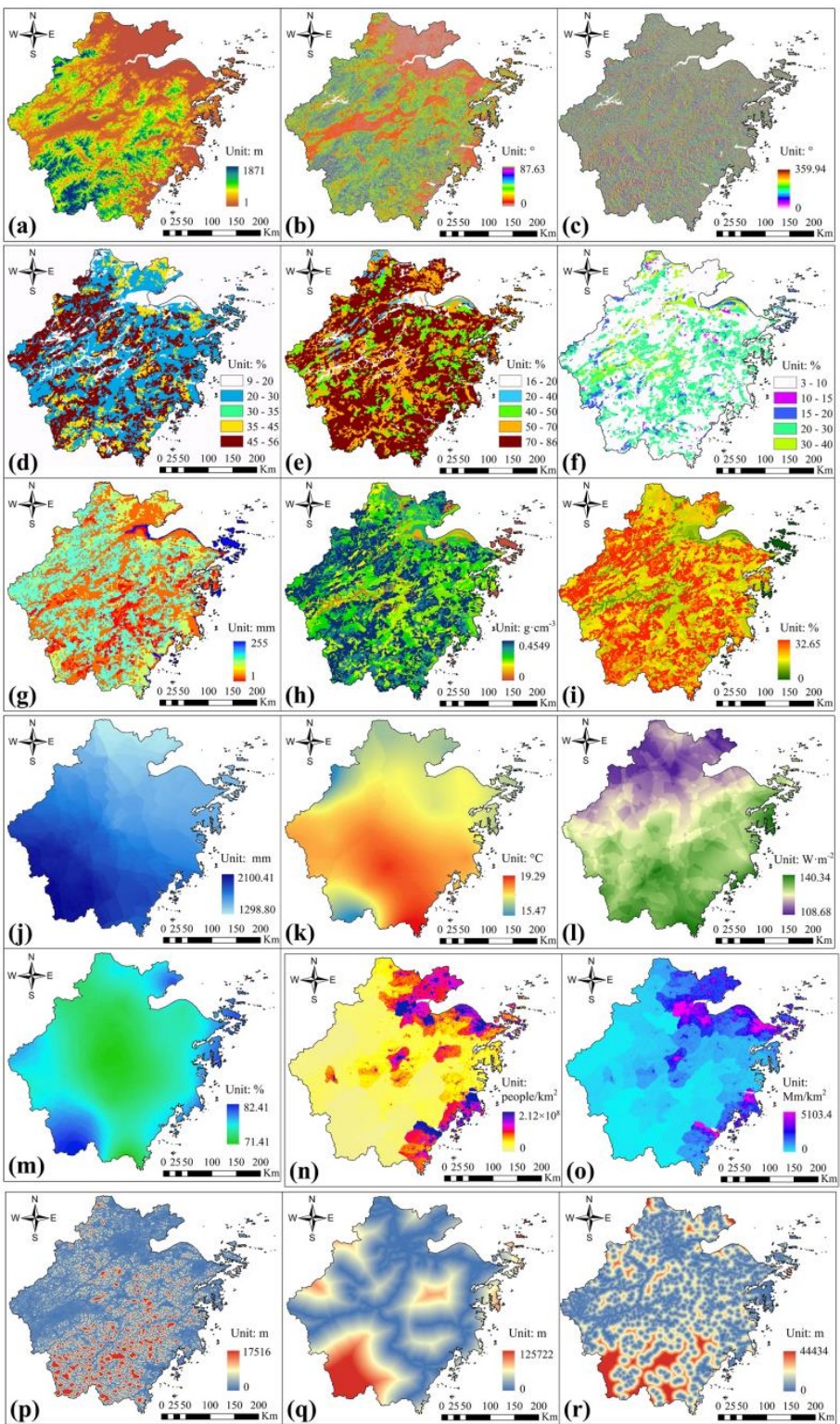

**Figure 3.** Topographic data in 2014: (**a**) DEM, (**b**) slope, and (**c**) aspect; soil data in 2008: (**d**) clay fraction, (**e**) sand fraction, (**f**) silt fraction, (**g**) soil available water content, (**h**) soil bulk density, and (**i**) soil wilt point; climate data in 2014: (**j**) total precipitation, (**k**) average temperature, (**l**) average radiation, and (**m**) average relative humidity; socioeconomic data in 2015: (**n**) population density, (**o**) GDP; distance data in 2014: (**p**) distance to road, (**q**) distance to railway, and (**r**) distance to water.

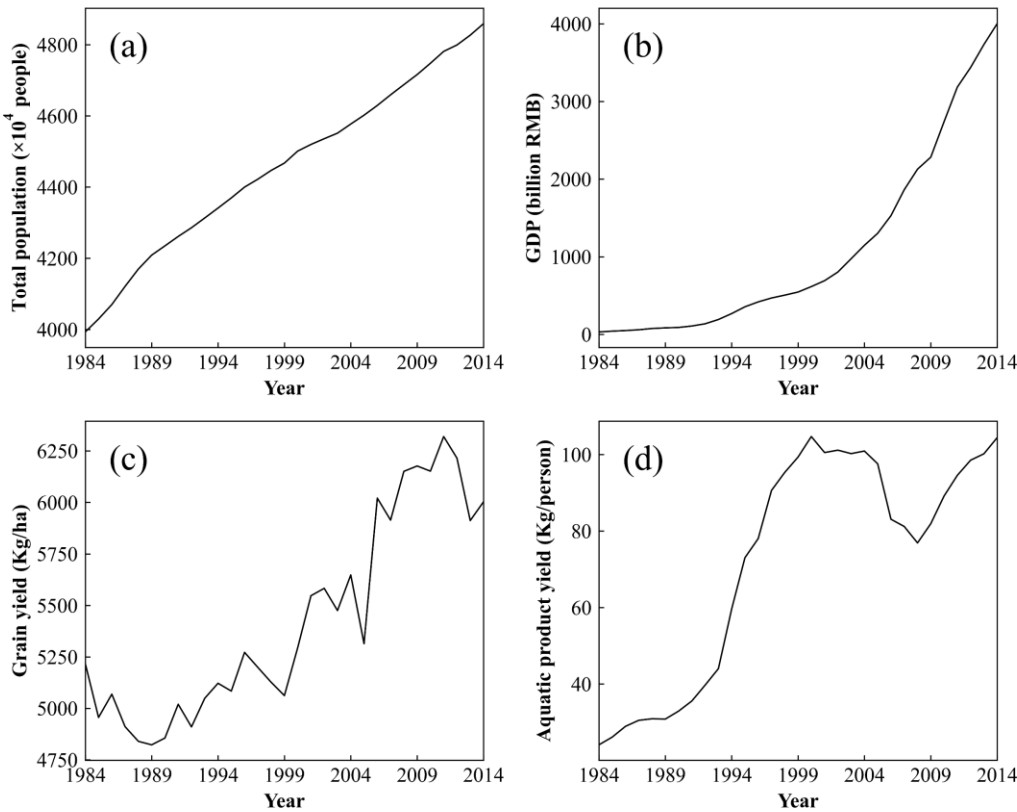

**Figure 4.** Macro statistics: (**a**) total population; (**b**) GDP; (**c**) grain yield; (**d**) aquatic product yield.

**Table 2.** The number of verification plots in different years.

| Year | UL | WB | CL | BLF | CF | BF | Total |
|------|-----|-----|-----|-----|-----|-----|-------|
| 1984 | 151 | 141 | 302 | 204 | 385 | 114 | 1297 |
| 1988 | 157 | 104 | 287 | 164 | 317 | 149 | 1178 |
| 1992 | 163 | 112 | 237 | 196 | 266 | 182 | 1156 |
| 1996 | 177 | 134 | 267 | 144 | 215 | 170 | 1107 |
| 2000 | 128 | 146 | 139 | 159 | 165 | 232 | 969 |
| 2004 | 128 | 142 | 139 | 142 | 152 | 215 | 918 |
| 2008 | 123 | 128 | 127 | 127 | 127 | 246 | 878 |
| 2014 | 123 | 132 | 138 | 147 | 154 | 139 | 833 |

*2.3. Future Scenario Description*

Four development scenarios (Figure 5) were designed based on the CMIP5 while considering climate variations together with different socioeconomic developments in Zhejiang Province.

Future climate data were derived from BCC-CSM1-1 climate change modeling data under the RCP 2.6, 4.5, 6.0, and 8.5 scenarios proposed by the IPCC AR5 [45]. The data under each scenario included annual total precipitation, annual average temperature, annual average radiation, and annual average relative humidity, which correspond to past climate data so that the BCS model can alternate input data in the prediction phase. Future socioeconomic scenario settings (Table 3) were mainly based on past macro-statistical data (Figure 4).

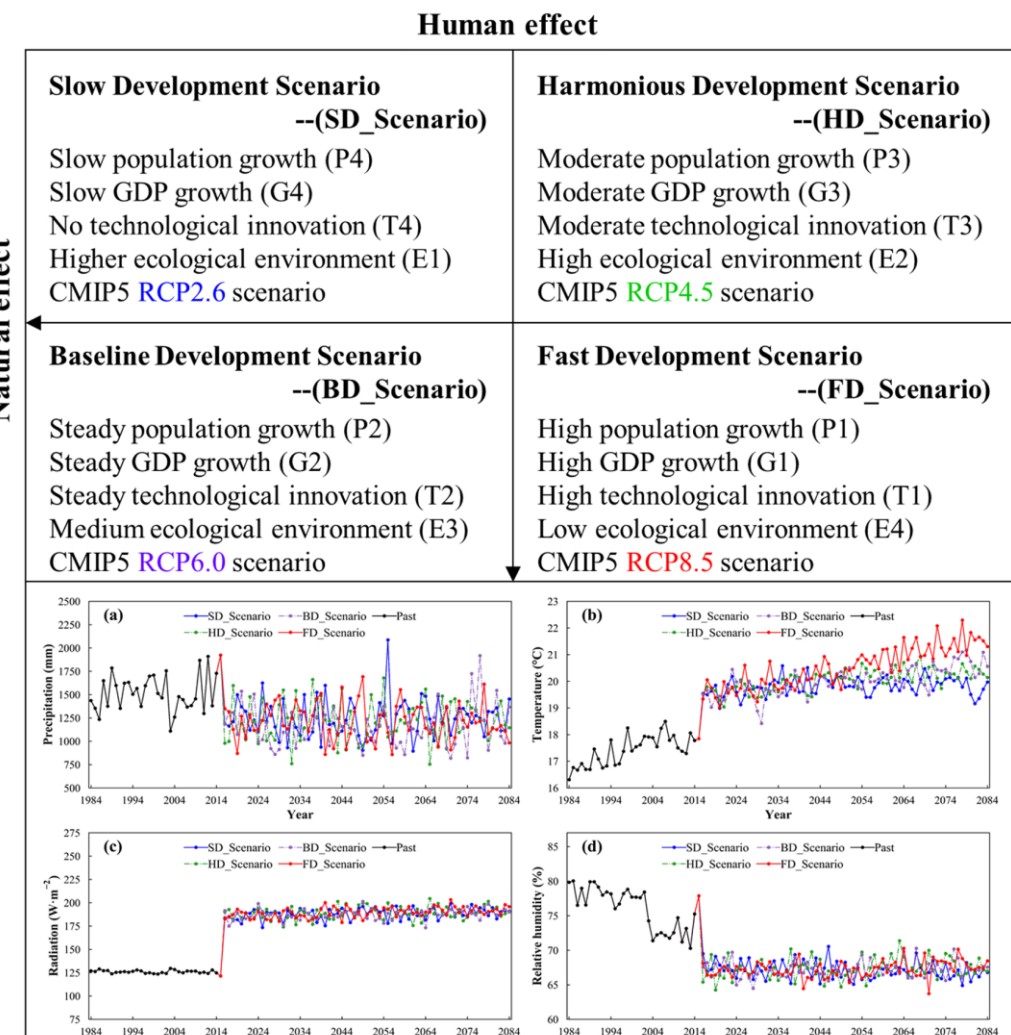

**Figure 5.** Configurations of four development scenarios concerning human and natural effects: (**a**) annual total precipitation; (**b**) annual average temperature; (**c**) annual average radiation; (**d**) annual average relative humidity under four scenarios.

**Table 3.** Parameter settings of different socioeconomic development scenarios.

| Factors | Patterns | Annual Growth Rate Settings from 2014 to 2084 |
|---|---|---|
| Population | High growth (P1) | 7.2‰ average from 2004 to 2014 |
| | Steady growth (P2) | Growth rate simulated by logistic population retardation growth model |
| | Moderate growth (P3) | 0.85× growth rate simulated by logistic population retardation growth model |
| | Slow growth (P4) | 7.2‰ linearly down to 3.4‰ |
| GDP | High growth (G1) | 14% average from 2004 to 2014 |
| | Steady growth (G2) | 14% linearly down to 10.5% |
| | Moderate growth (G3) | 14% linearly down to 8% |
| | Slow growth (G4) | 14% linearly down to 6.5% |

**Table 3.** *Cont.*

| Factors | Patterns | | Annual Growth Rate Settings from 2014 to 2084 |
|---|---|---|---|
| Technology | Rapid innovation (T1) | Grain yield | Maintain 5‰ in 2014 |
| | | Aquatic yield | Maintain 8% in 2014 |
| | Steady innovation (T2) | Grain yield | 5‰ linearly down to 3‰ |
| | | Aquatic yield | 8% linearly down to 5% |
| | Moderate innovation (T3) | Grain yield | 5% linearly down to 1% |
| | | Aquatic yield | 8% linearly down to 2% |
| | No innovation (T4) | Grain yield | 0% |
| | | Aquatic yield | 0% |
| Ecology | Higher forest coverage rate (E1) | | 60.89% linearly up to 65% |
| | High forest coverage rate (E2) | | 60.89% linearly up to 63% |
| | Medium forest coverage rate (E3) | | 60.89% linearly up to 61% |
| | Low forest coverage rate (E4) | | 60.89% linearly down to 60% |

The logistic population retardation growth model [46] was constructed to calculate $p_m$ and $r$. $p_m$ is the maximum population of 4,328,457, and $r$ is the population growth rate of 6.097%.

The slow development scenario (SD_Scenario) was constructed to predict the demand for land use under the influence of slow socioeconomic growth and mild climate change. Corresponding to the RCP2.6 scenario, it is the lowest scenario for greenhouse gas emissions and radiative forcing. Radiative forcing will increase first and decrease after 2054, reaching 2.6 $W \cdot m^{-2}$ by 2100, with global average warming limited to 2 °C [47]. In this scenario, the population, GDP, and technological innovation are considered to be at their lowest levels, while the annual temperature and precipitation change slowly. It promotes the use of biomass energy and advocates the restoration of forests, so the forests will be well restored and the forest coverage rate will be the highest.

The harmonious development scenario (HD_Scenario) is a sustainable development model. Moderate growth of the population and GDP are taken into account, and the proportion of investment is assumed to have more input into the productivity of agriculture and fisheries. Additionally, the natural environment will undergo moderate changes (RCP4.5 scenario), forming a model of sustainable development with the economy and society. This scenario limits greenhouse gas emissions through low-emission energy technologies so that increasing radiative forcing will only reach 4.5 $W \cdot m^{-2}$ by 2100.

The baseline development scenario (BD_Scenario) is established based on the past and current development trends of Zhejiang Province. Under this scenario, the population, economy, and technological level are recognized as stable and advanced. Moreover, the climate is assumed to maintain its current temperature and precipitation rates, which is consistent with the RCP6.0 scenario [48].

Contrary to the SD_Scenario, the fast development scenario (FD_Scenario) aims to maximize the social and economic benefits of Zhejiang Province. The economy and population increase at a high speed, and science and technology develop rapidly. At the same time, massive-scale human activities will accelerate greenhouse gas emissions and increase atmospheric radiation, leading to a sharp increase in temperature (RCP 8.5) and, thus, resulting in drastic climate change [49,50].

*2.4. Methodology*

In the paper, we present the BCS model for multiple LUCC scenarios for future land use by coupling human and natural effects. The proposed model consists of an SD model and a BPNN_CA model. The SD model was used to project the land-use demand at the macro-level, and the BPNN_CA model was used to allocate land use spatially at the micro-level.

The flow chart used is shown in Figure 6. Firstly, based on the geospatial data in the past, sample sets were randomly sampled to train the BPNN, and then the overall probability was calculated, and the simulated land-use types were determined according to the overall probability and roulette-wheel selection mechanism. Secondly, macro-statistical

data were used to establish the relationship between land-use area and macro-factors so as to capture the feedback loop between them. Thirdly, the SD model was interactively coupled with the CA model, and the simulated land-use pattern was output only when the spatially allocated area of the CA model reached the requirements of the SD model. Finally, when the model accuracy was valid, the inputs of the two sub-models were updated using future data to obtain the land-use pattern in the future.

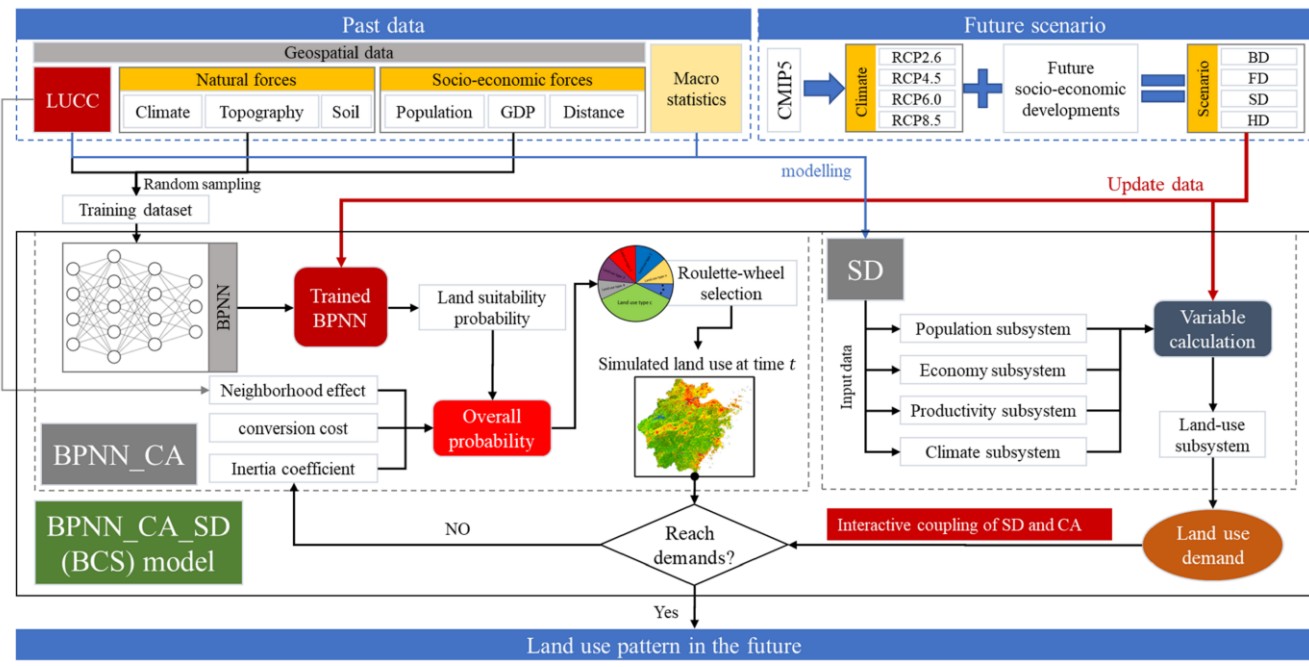

**Figure 6.** Flow chart used in this study.

### 2.4.1. SD Model

The SD model for LUCC was mainly used to describe the quantitative conversion between different land-use types to achieve quantitative demand forecasting [10,51]. The relationship of quantitative transformation requires consideration of the market, investment, policies, and climate adaptation strategies from a macro perspective [8]. By understanding the impact of the internal factors of the macro system on LUCC, a top-down quantitative model can be formed [12].

The SD model was built using the Vensim PLE software (Figure 7). The simulation period of the SD model was 1984–2084 and the time step of the model was one year. It included the following two stages: (1) 1984–2014 was the model test stage, and the historical data were used to set the parameters, adjust the model, and validate the model; (2) 2014–2084 was the prediction stage, and the future land-use demand under different scenarios could be simulated.

To simulate the quantitative demand by considering both human and natural effects, the SD model in our study consisted of the following five subsystems: population, economy, productivity, climate, and land use. As the main social factor, the population influences the land-use system in many aspects. The economy has strong influences on population and land use such as gross domestic product (GDP), which affects the change in fixed-asset investments, thereby driving economic investment in various land-use types. The productivity mainly includes food and aquatic products; the amount of productivity indirectly determines the increase and decrease in cultivated land and water bodies. The climate subsystem involves temperature and precipitation, which have various influences on the growth and regeneration capacity of vegetation, leading to changes in cultivated land and forest land. The land-use subsystem includes the increment or decrement of each land-use demand. The change in each land-use type is constrained by the integrated influences of

socioeconomic and climate conditions as well as by the interactions among the various land-use types.

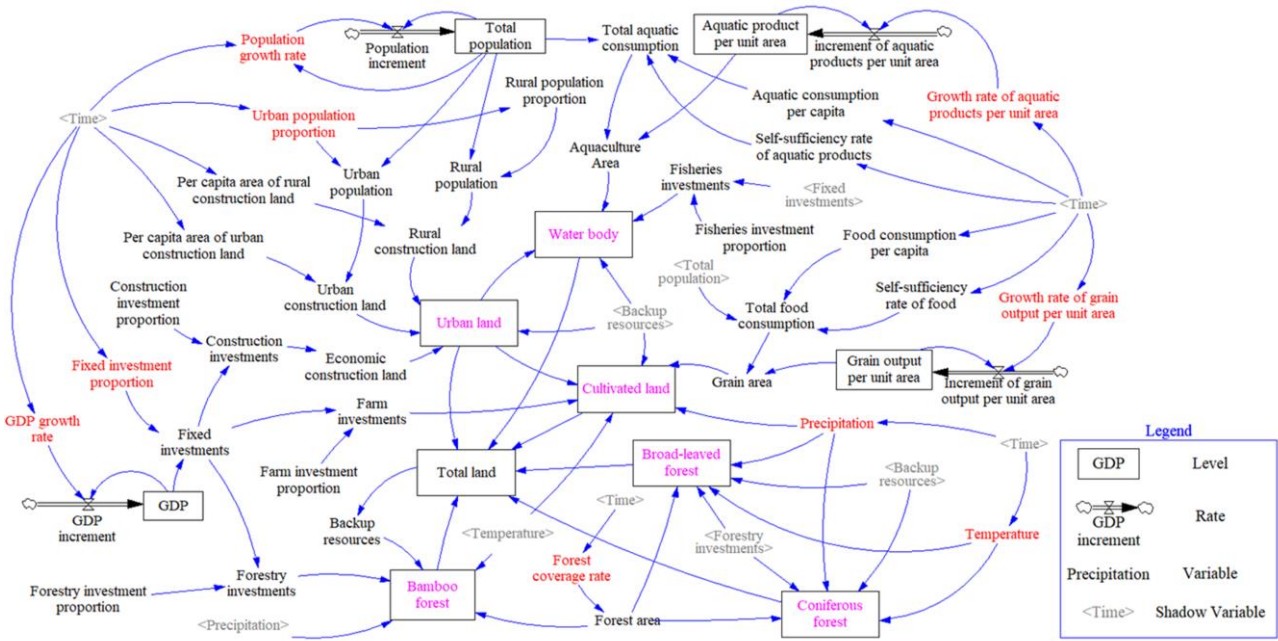

**Figure 7.** The structure of the SD model. Variables in red indicate the inputs of the model, which varied in different scenarios. Variables in purple represent the main outputs, including the areas of different LULC types. Purple ones were used to validate the model.

The values of the level variables (e.g., total population, GDP) for each year are obtained based on the annual growth rate for that year compared to the previous year. Under different development scenarios, the annual growth rates are set differently, and the land-use demand outputs obtained by relevant formulas will also be greatly different.

### 2.4.2. BPNN_CA Model

The CA model is the most common and widely used model for LUCC spatial simulations. Based on the Python language programming, the CA model can be integrated with BPNN as the BPNN_CA model. Whether a cell state changes is not only related to external factors (land suitability) but also depends on the interrelationships between various land uses (e.g., neighborhood effects, land inheritance) and stochastic occurrences [26]. Therefore, the transition rules of the model can be expressed as follows:

$$State_{i,t+1} = f\left(Pro_{i,k}^t\right) \tag{1}$$

where $State_{i,t+1}$ is the land-use type of a cell $i$ at time $t + 1$. $f$ is the land-use transition rule that decides the change of a cell $i$ from the state ($State_{i,t}$) at time $t$ to the state ($State_{i,t+1}$) at time $t + 1$. $Pro_{i,k}^t$ is the overall probability of the cell $i$ being occupied by a specific land-use type $k$ at time $t$, which is defined as follows:

$$Pro_{i,k}^t = Ls_{i,k} \times \Omega_{i,k}^t \times I_k^t \times (1 - C_{s \rightarrow k}) \tag{2}$$

(1) $Ls_{i,k}$ is the land suitability probability of a cell $i$ being occupied by land-use type $k$, which represents the impact of the driving factors on land-use transition. The collected driving factors (Figure 3) normalized to the range of [0,1] were selected to establish the BPNN model. The BPNN had the following four layers: an input layer, two hidden layers, and an output layer, with the corresponding numbers of neurons set to (19, 30, 14, 6). Simultaneously, the sigmoid function was selected as the activation function to ensure that

$Ls_{i,k}$ fell within [0,1]. A total of 6 million randomly selected samples were divided into a training set and a test set in a ratio of 7:3. The training set was used to train the neural network, while the test set was used to evaluate its performance. The minibatch gradient descent method (Algorithm 1) was adopted to input the training set into the BPNN module in batches to train and optimize the parameters of the network. In the algorithm, the number of iterations ($K$) was 200, and the initial learning rate was 0.0005. In each iteration, the loss function of the mean square error ($MSE$) was used to compute the gradients ($grad$), and the adaptive optimization of Adam was used to automatically adjust the learning rate ($lr$) after training a batch of images. The training of the BPNN stopped after $K$ passed through the data set.

---

**Algorithm 1: Train BPNN with the minibatch Adam optimization algorithm.**

---

initialize ($net$)
**for** $epoch = 1, \ldots, K$ **do**
    **for** $batch = 1, \ldots, \# images/b$ **do**
        $images \leftarrow$ uniformly sample $batch - size$ images
        $X, y \leftarrow$ preprocess(images)
        $Ls_{i,k} \leftarrow$ forward (net, $X$)
        $MSE \leftarrow$ loss ($Ls_{i,k}, y$)
        $lr, grad \leftarrow$ backpropagation ($MSE$)
        update ($net, lr, grad$)
    **end for**
**end for**

---

(2) $\Omega_{i,k}^{t}$ is the neighborhood effect of land-use type $k$ on grid cell $i$ at iteration time $t$ affected by the surrounding neighborhood at iteration time $t - 1$, which is given as follows:

$$\Omega_{i,k}^{t} = \frac{\sum_{N \times N}(State_{i,t-1} = k)}{N \times N - 1} \tag{3}$$

where $\sum_{N \times N}(State_{i,t-1} = k)$ is the total number of grid cells occupied by land-use type $k$ at iteration time $t - 1$ within the $N \times N$ window. In this study, the $5 \times 5$ Moore neighborhood was adopted.

(3) $I_{k}^{t}$ is the inertia coefficient of land-use type $k$ at iteration time $t$. If the development of a specific land-use type contradicts the future quantitative demands, the inertia coefficient would dynamically control the inheritance of the land-use type to increase or decrease to rectify the changing trend in the next iteration, which is expressed as follows:

$$I_{k}^{t} = \begin{cases} I_{k}^{t-1} & , \quad if \left| D_{k}^{t-1} \right| \leqq \left| D_{k}^{t-2} \right| \\ I_{k}^{t-1} \times \frac{D_{k}^{t-2}}{D_{k}^{t-1}} & , \quad if D_{k}^{t-1} < D_{k}^{t-2} < 0 \\ I_{k}^{t-1} \times \frac{D_{k}^{t-1}}{D_{k}^{t-2}} & , \quad if 0 < D_{k}^{t-2} < D_{k}^{t-1} \end{cases} \tag{4}$$

where $D_{k}^{t-1} D_{k}^{t-2}$ are the differences between the quantitative demand and the allocated amount of land-use type $k$ until the iteration times of $t - 1$ and $t - 2$, respectively. Based on the ratio of the two values, the model updates $I_{k}^{t}$ in real time.

(4) $C_{s \rightarrow k}$ is the conversion cost, which is the difficulty of converting a specific grid pixel from land-use type $s$ to the target land-use type $k$. It is a constant parameter, the value of which is fixed within [0,1] (Table 4). Additionally, larger values indicate more difficult conversions.

**Table 4.** The conversion cost of different land-use types from 2004 to 2084.

| Types | UL | WB | CL | BLF | CF | BF |
|-------|------|------|------|------|------|------|
| UL | 0 | 0.85 | 0.7 | 0.99 | 0.99 | 0.99 |
| WB | 0.8 | 0 | 0.8 | 0.9 | 0.9 | 0.9 |
| CL | 0.3 | 0.7 | 0 | 0.5 | 0.4 | 0.5 |
| BLF | 0.9 | 0.9 | 0.7 | 0 | 0.6 | 0.5 |
| CF | 0.9 | 0.9 | 0.6 | 0.5 | 0 | 0.6 |
| BF | 0.9 | 0.9 | 0.7 | 0.6 | 0.6 | 0 |

(5) The $f$ for this study was a roulette-wheel selection mechanism [52], which assumes that the probability of selection is proportional to the fitness of a sector. Let us consider $N$ sectors, each characterized by its fitness $p_{i,k}^t > 0$ ($n = 1, \ldots, N$). As shown in Algorithm 2, one constructs a line segment of length 1 out of consecutive sectors of length $p_{i,k}^t$, generates a uniformly distributed random number $r_i^t$ such that $r_i^t \leq q_{i,n}^t$, and locates the corresponding sector, thus selecting the respective label $Type_n$ to assign to the state ($State_{i,t+1}$) at time $t + 1$. In this paper, $n = k = 1, \ldots, 6$ and $Type_n$ = (UL, WB, CL, BLF, CF, BF). Spatial allocation using this mechanism not only ensures the possibility of all events occurring but also ensures the dominance of high probability events, therefore ensuring the randomness and fairness in the allocation and effectively reducing the uncertainty of LUCC.

---

**Algorithm 2: Using a roulette-wheel selection mechanism to allocate the probability.**

---

**input:** $Pro_{i,k}^t$
$p_{i,k}^t \leftarrow Pro_{i,k}^t / \sum_{n=1}^{N} Pro_{i,k}^t$
$q_{i,n}^t \leftarrow \sum_{k=1}^{n} p_{i,k}^t$
$r_i^t \leftarrow$ a uniformly distributed random number ranging from 0 to 1
**for** $n = 1, \ldots, N$ **do**
    **if** $r_i^t \leq q_{i,n}^t$ **then**
        $State_{i,t+1} \leftarrow Type_n$
        **break**
    **else**
        **continue**
**end for**

---

### 2.4.3. Interactive Integration of the BCS Model

Many integrated models are loosely coupled with the quantitative model and the spatial model based on the final land-use demands so that the accuracy is not as high as with the interactive coupling model [9]. Thus, to strengthen the mutual feedback between the SD and BPNN_CA sub-models, the two sub-models in the BCS model are interactively coupled during the study time series.

The schematic diagram of the coupling mechanism is shown in Figure 8. The future 70 year period from 2014 to 2084 was divided into the following 14 segments at 5 year intervals: 2014–2019, 2019–2024, ..., and 2079–2084. Using the actual classification in 2014 as input data, the process was as follows: (1) the land-use pattern was input to the BPNN_CA to calculate land suitability and neighborhood effects, etc., while the area of each land-use type was fed into the SD model; (2) the demand of each land-use type at the next time was predicted through the configured SD model using the area at the previous time combined with the effects of human and natural factors during this time interval; (3) the demand was input to the BPNN_CA model as a constraint at the end of the model iteration, and the CA model explored the local competition and interactions by adjusting the $I_k^t$ through multiple iterations; (4) the spatial distribution of the land-use pattern for the next moment was generated until the area allocated by the BPNN_CA model met the demand of the SD model. Looping steps (1)–(4) above, finally, the SD and BPNN_CA models would exchange input/output information to generate the land-use pattern in 2019, 2024, ..., 2084 sequentially.

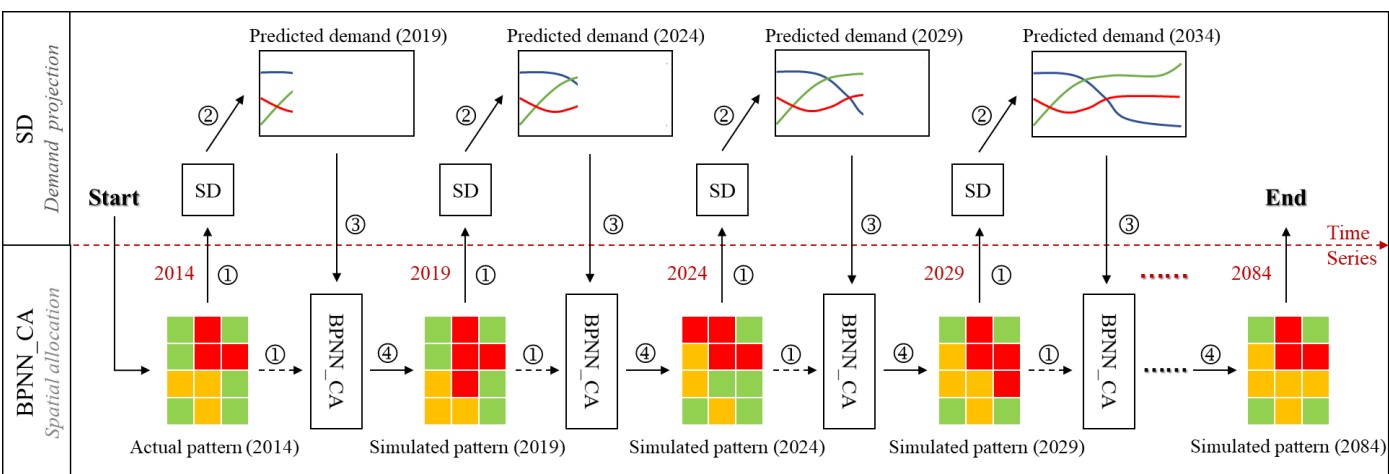

**Figure 8.** The interactive coupling mechanism of the SD model and BPNN_CA model.

2.4.4. Assessment Methods of the BCS Model

The performance evaluation of the BCS coupled model needs to verify its ability to simulate the area and spatial pattern of land use.

For the area simulation, the determination coefficient $(R^2)$ and root mean square error (RMSE) were calculated using predicted area and actual area to account for the fitness of the SD model. The higher the $R^2$ and the smaller the RMSE, the better the result. The $R^2$ and RMSE are defined as follows:

$$R^2 = 1 - \frac{\sum_{i=1}^{N}(y_p - y_a)_i^2}{\sum_{i=1}^{N}(y_p - \overline{y}_a)_i^2} \tag{5}$$

$$RMSE = \sqrt{\frac{1}{N}\sum_{i=1}^{N}|y_p - \overline{y}_a|_i^2} \tag{6}$$

where $y_p$ represents the simulated area, $y_a$ represents the actual area, $\overline{y}_a$ represents the average actual area, and $N$ represents the number of land-use types, $N = 6$.

For the spatial simulation, not only were OA, Kappa, and PA used for the evaluation but also a figure of merit (FOM) was introduced. The reason was that OA can only show the overall consistency of the simulation results and the actual results, but it cannot show the consistency of cell state changes, and the *FoM* index can directly show the ability of the model to simulate changes [53]. The OA, Kappa, PA, and FOM are expressed as follows:

$$OA = a_0 = \frac{1}{n}\sum_{k=1}^{N}a_{kk} \tag{7}$$

$$Kappa = \frac{a_0 - a_c}{1 - a_c}, \ a_c = \frac{\sum_{k=1}^{N}\left(\sum_{i=1}^{N}a_{ki}*t_k\right)}{n*n} \tag{8}$$

$$PA_k = a_{kk}/t_k \tag{9}$$

$$FOM = Rights/(Rights + Misses + Faults + Extras) \tag{10}$$

where $n$ represents the total number of pixels; $k$ and $i$ represents the land-use type, $k, i = 1, 2, 3 \ldots N$; $a_{kk}$ refers to the number of the category $k$ correctly simulated; $a_{ki}$ is the number of type $i$ simulated as $k$; $t_k$ refers to the actual total number of category $k$; *Misses* is an area of error due to the actual changes simulated as no change; *Rights* is an area of correctness due to the actual changes consistent with the simulated changes; *Faults* is an area of error

due to the actual change simulated as changing to an incorrect category; *Extras* is an area of error due to the fact of no actual change simulated as change.

## 3. Results

### 3.1. Model Validations

The simulation for 2014 obtained through the BCS model is shown in Figure 9. Two partial enlargements not only show that the result was consistent with the actual pattern but also indicate that the spatial LUCC from 2008–2014 was also well simulated. However, there were still some errors in local details, and not all changes could be simulated; therefore, we quantitatively evaluated them to determine the performance of the BCS model.

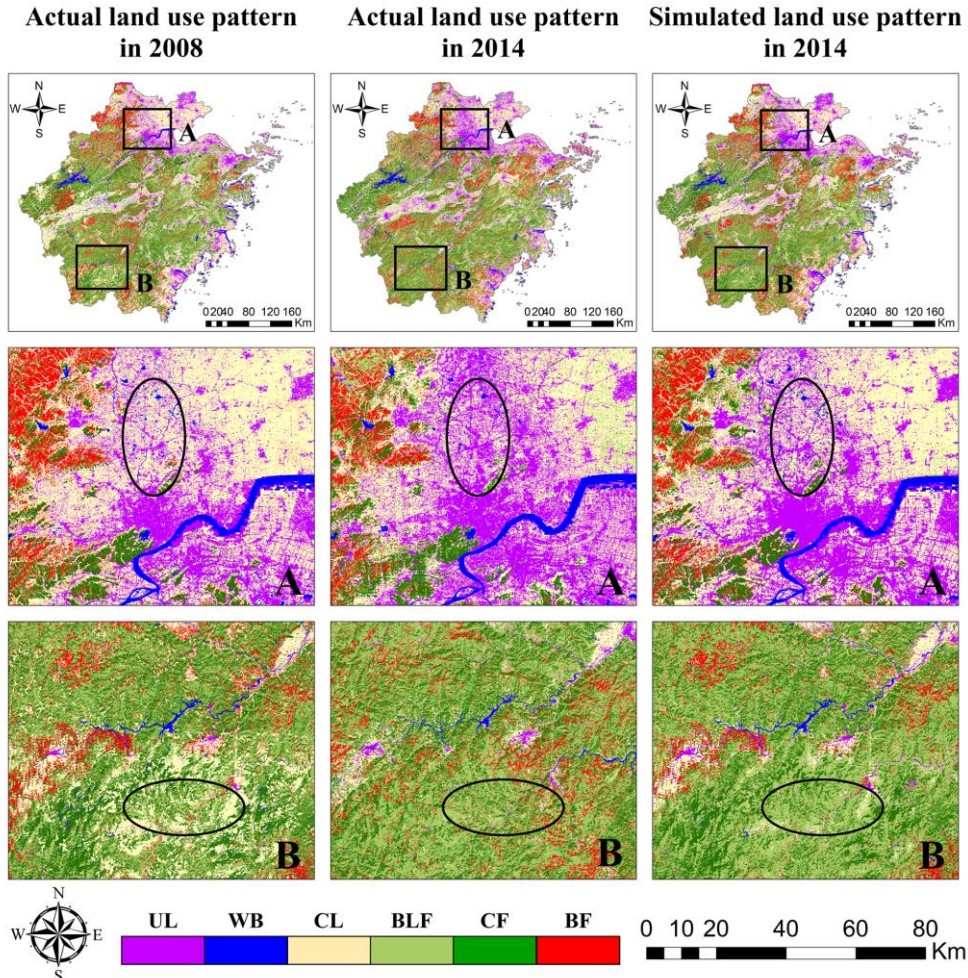

**Figure 9.** The actual land-use pattern in 2008 and 2014 and the simulated land-use pattern in 2014. (**A**,**B**) are the two sub-areas shown in magnification.

Based on the simulation results and the actual classification in 2014, we evaluated the BCS model quantitatively, which consisted of the following four components: (1) the $R^2$ and RMSE were used to evaluate the SD model for predicting land-use area. The high $R^2$ (0.99) with the small RMSE (707 km$^2$) demonstrates that the calibrated SD model provides an important basis for the accurate prediction of future land-use area demand (Figure 10a); (2) The receiver operating characteristic (ROC) curve and the area under the ROC curve (AUC) values were used to quantify the BPNN's performance. A larger AUC value corresponds to a better BPNN fitting performance [17]. The AUC values were all above 0.8 (Figure 10b), indicating that the land suitability fit for each land use can be strongly explained by the selected driving factors; (3) The normalized confusion matrix (Figure 10c) was used to evaluate the spatial consistency of the two maps in 2014. The high OA (79.61%)

and Kappa (75.53%) show that the BCS model simulation results in 2014 were in excellent spatial consistency with the actual classification; (4) The FOM calculated from the simulation results in 2014 and the actual classifications in 2008 and 2014 was 28.21%, representing a large proportion of the area where the actual change was consistent with the simulated change (Figure 10d). Pontius et al. [54] reported FOM ranging from 1 to 59%, most of which were lower than 30%. Considering that our study area was a large region with complex climatic conditions and significant regional differences, the simulation accuracy was quite acceptable.

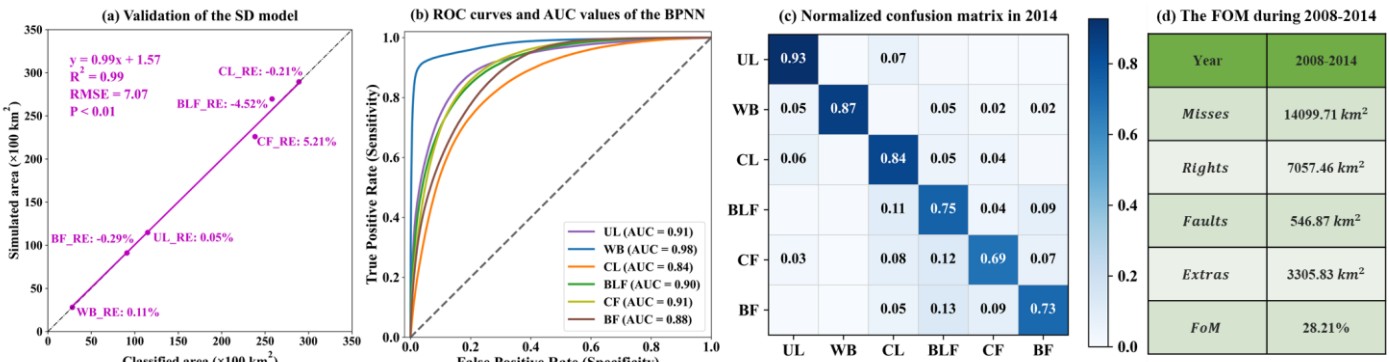

**Figure 10.** Verification: (**a**) comparison between the classified and simulated areas using the SD model; (**b**) ROC curves and AUC values fitted by the BPNN; (**c**) normalized confusion matrix of the simulated result and the actual classification in 2014; (**d**) the FOM from 2008 to 2014.

In summary, the BCS coupled model can not only be used to simulate great land-use spatial distributions but can also be used to show consistent LUCC between the simulated results and the actual classification, guaranteeing an accurate simulation of the future spatial distribution of land use.

### 3.2. Future Land Use Demand Projection

According to the four scenarios described in Section 2.3, the SD model with the interaction mechanism was used to predict the land-use demands for the next 70 years (Figure 11).

Figure 11a shows that the UL area will increase constantly from 2014 to 2084 under all scenarios due to the population increase. In the SD_Scenario and FD_Scenario, it will grow to $1.28 \times 10^4$ km$^2$ and $1.63 \times 10^4$ km$^2$ in 2084. Compared to the area in 2014, it will increase by 10% and 15% (+10% and +15%), respectively. The area in the BD_Scenario and HD_Scenario is in between that in the first two scenarios.

Figure 11b shows that there will be slight WB area changes in the future. It is between 2250 and 2750 km$^2$ in the next 70 years under each scenario.

Figure 11c shows that the CL area will gradually decrease in the future. In the SD_Scenario and FD_Scenario, it will be reduced to $2.08 \times 10^4$ and $2.37 \times 10^4$ km$^2$ in 2084 (–28% and –18%), respectively. The area in the BD_Scenario and HD_Scenario was in between that in the first two scenarios.

Figure 11d shows the BF area has slowly increased since 1984. Its differences between different scenarios were small in 2014–2054, while the differences started to become apparent after 2054. By 2084, in the FD_Scenario, it will reach $1.25 \times 10^4$ km$^2$ (+37%), which is 1217 km$^2$ more than that in the SD_Scenario.

Figure 11e shows the BLF areas have increased obviously over the past 30 years, while in the future 70 years, they will fluctuate between $2.25 \times 10^4$ km$^2$ and $3 \times 10^4$ km$^2$, with minor differences and changes among different scenarios.

Figure 11f shows the CF area varied clearly under different scenarios. In the FD_Scenario, it will have a decreasing trend after 2019, decreasing to $2.04 \times 10^4$ km$^2$ in 2084, indicating that the urbanization and other development in Zhejiang Province under this scenario will have a strong negative effect on the development of the CF area. While in the SD_Scenario, it will

reach $2.72 \times 10^4$ km$^2$ (+13%), revealing that the ecological focus and better protection of CF under this scenario has led to an increase rather than a decrease in CF area.

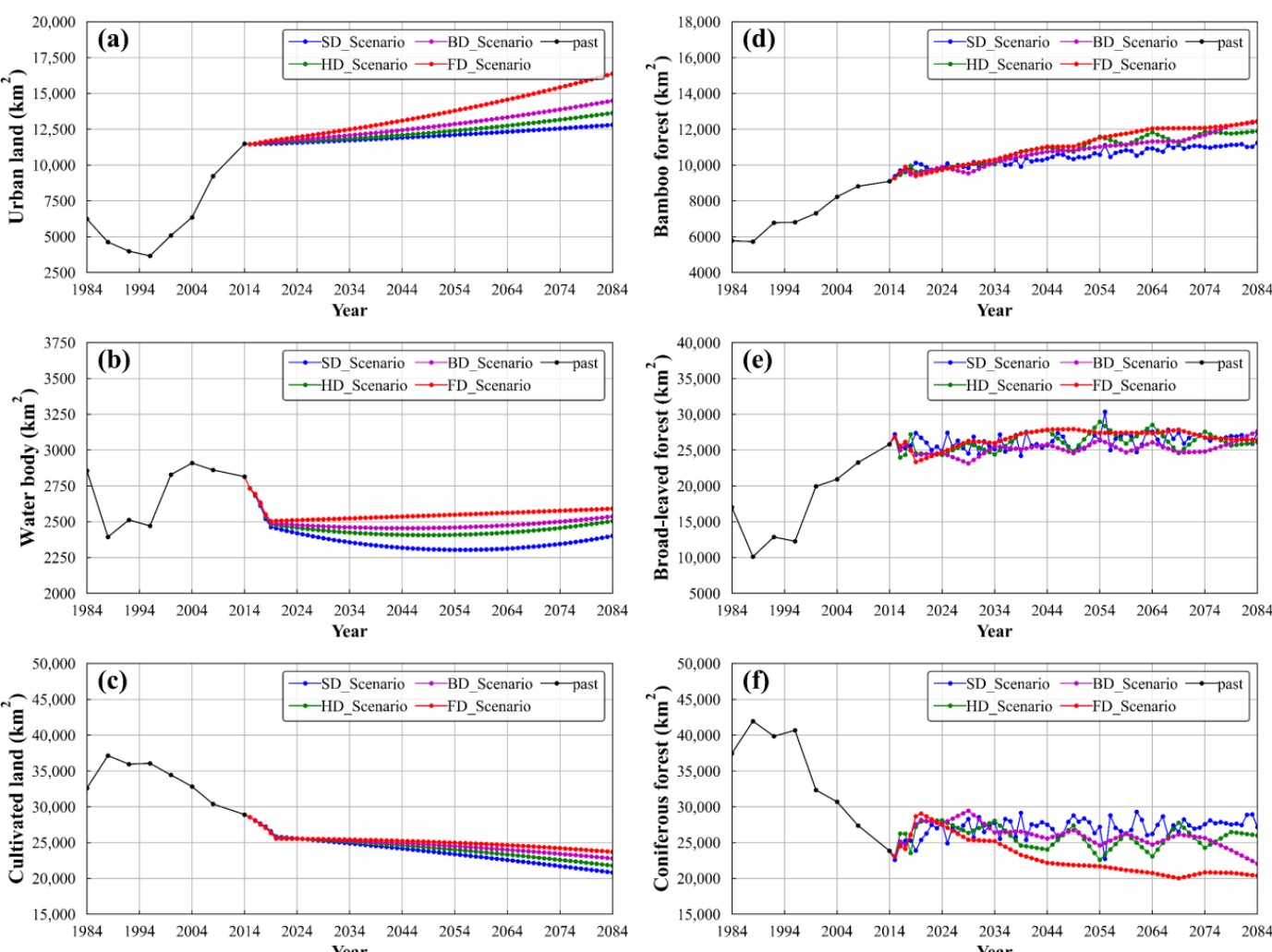

**Figure 11.** Areas of each land-use type in Zhejiang Province over the past 30 years and that over the next 70 years under different scenarios: (**a**) urban land area; (**b**) water body area; (**c**) cultivated land area; (**d**) bamboo forest area; (**e**) broad-leaved forest area; (**f**) coniferous forest area.

### 3.3. Future Spatiotemporal Land-Use Pattern

Based on the BCS coupled model, future land-use patterns in 2084 were simulated. To demonstrate the differences between different scenarios, two enlargements of the simulation results are shown in Figure 12.

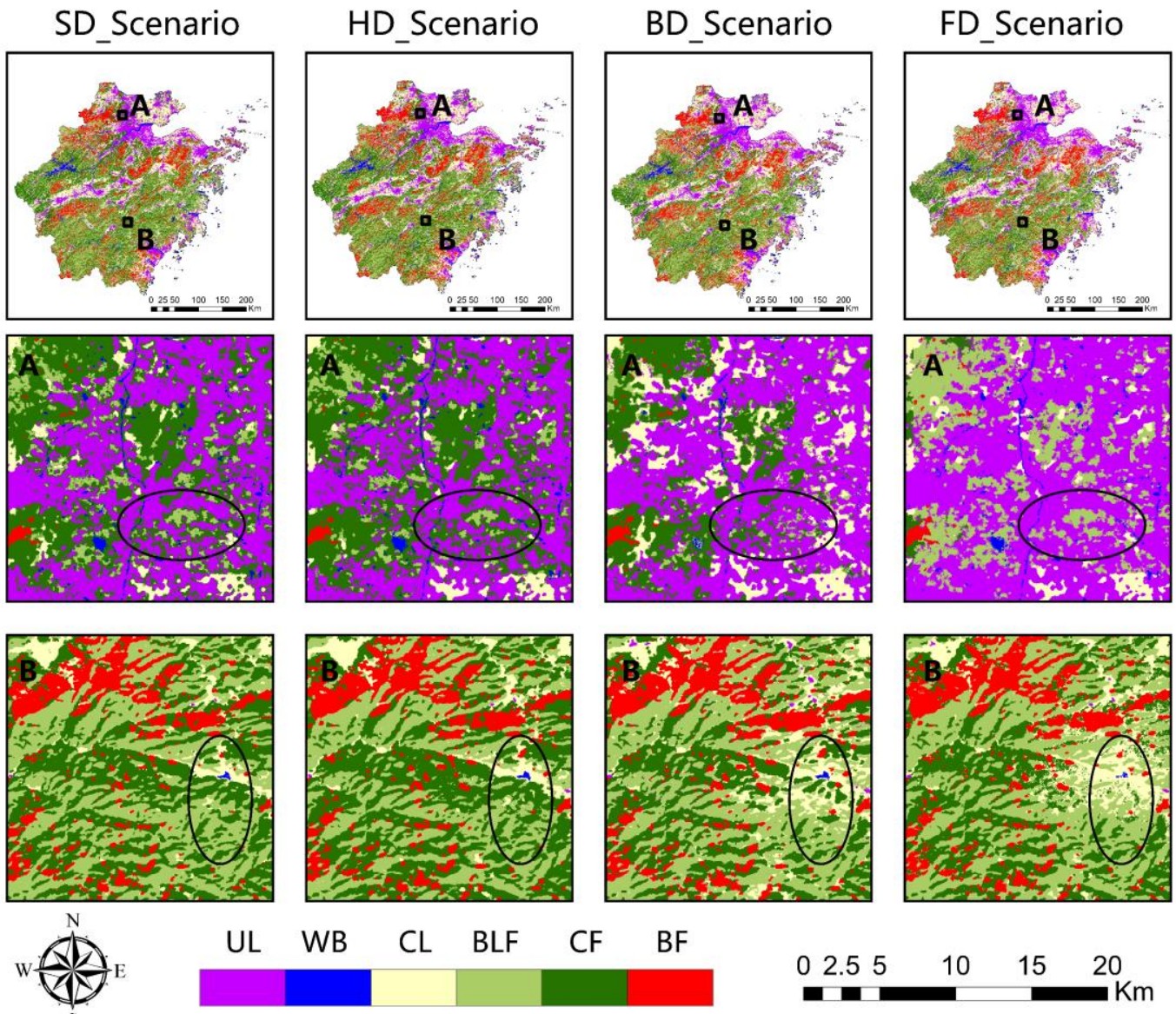

**Figure 12.** Spatial distribution of land use in different regions of Zhejiang Province in 2084 under different scenarios. (**A**,**B**) are the two sub-areas shown in magnification.

### 3.4. Analysis of Future Land-Use Conversion

With a total of six land-use types in this study, there were 36 possibilities for future land conversion. The Sankey diagrams (Figure 13) were used to show the land-use conversion from 2014 to 2084 under four scenarios, and the top five highest change amplitudes of all land uses are marked on the diagrams.

In both the SD_Scenario and HD_Scenario, the land-use types with the largest area outflow is BLF, with $4.8 \times 10^3$ km$^2$ (18.61%) and $6 \times 10^3$ km$^2$ (23.37%) of BLF will convert to CF in 2084 as a result of different climate changes and anthropogenic disturbances, respectively, and therefore the CF area will increase under both scenarios (Figure 13a,b). However, in the BD_Scenario and FD_Scenario, the percentage of CL outflow was both the highest due to the rapid urban development, and $5.2 \times 10^3$ km$^2$ (18.11%) and $5.7 \times 10^3$ km$^2$ (19.74%) of CL will convert to UL, respectively. These demonstrate a contrast between the ecological conservation scenarios and the urban development scenarios.

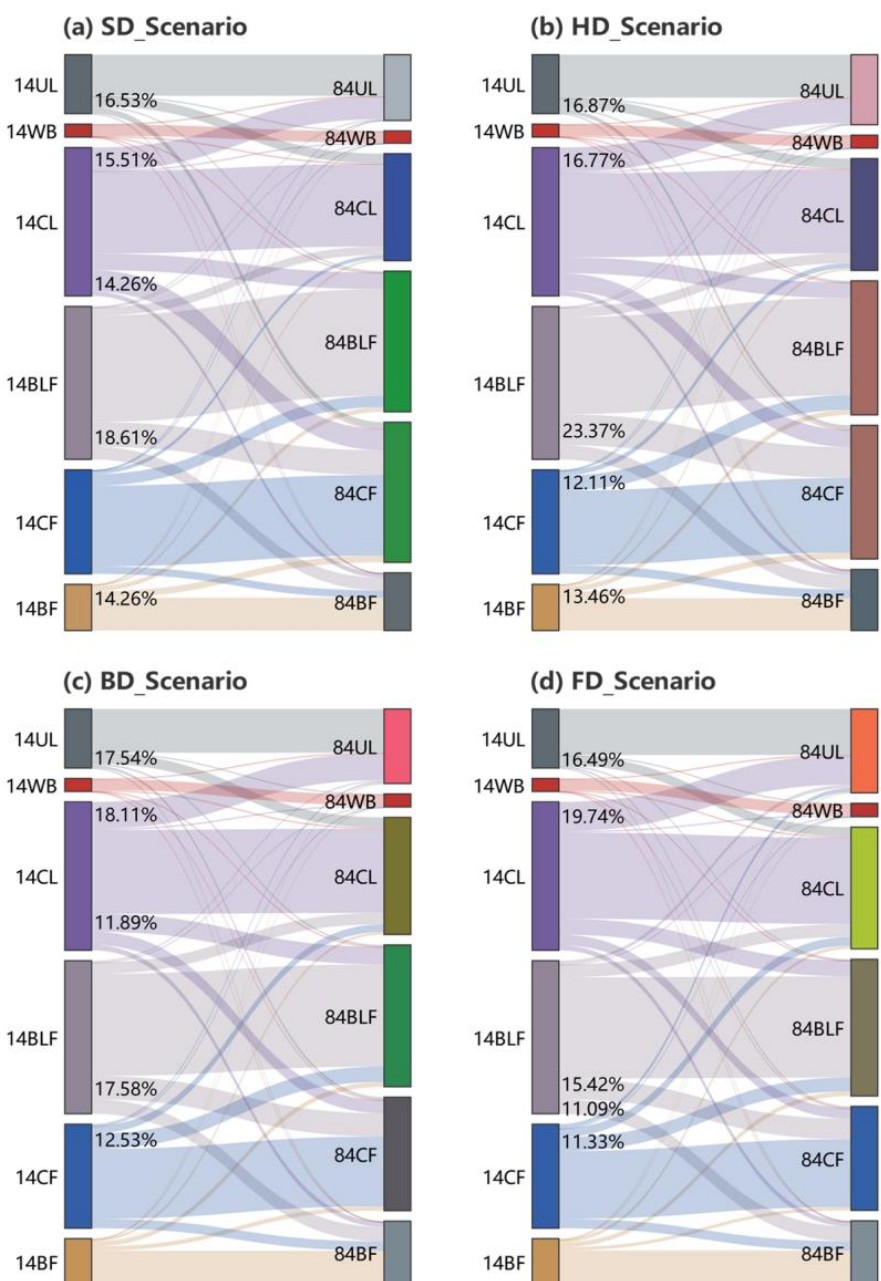

**Figure 13.** Predicted land-use conversion in 2014–2084 under 4 scenarios: (**a**) SD_Scenario; (**b**) HD_Scenario; (**c**) BD_Scenario; (**d**) FD_Scenario. Numbers: years; UL: urban land; WB: water body; CL: cultivated land; BLF: broad-leaved forest; CF: coniferous forest; BF: bamboo forest. For example, 14BF represents bamboo forest in 2014.

### 3.5. Analysis of Land-Use Change Amplitude at the Administrative Level

Based on the land-use pattern between 2014 and 2084, under different scenarios, the change amplitude of each land-use type at the Zhejiang Province administrative level was mapped as shown in Figure 14. The following three points are worth noting: (1) The BF area is in a state of growth in most of prefectures (+20% to +60%) under each scenario; (2) under the FD_Scenario, the CF area in northeast Zhejiang will be greatly reduced (–20% to –95%); (3) the BLF area in Jiaxing will decrease under either scenario (–24% to –90%).

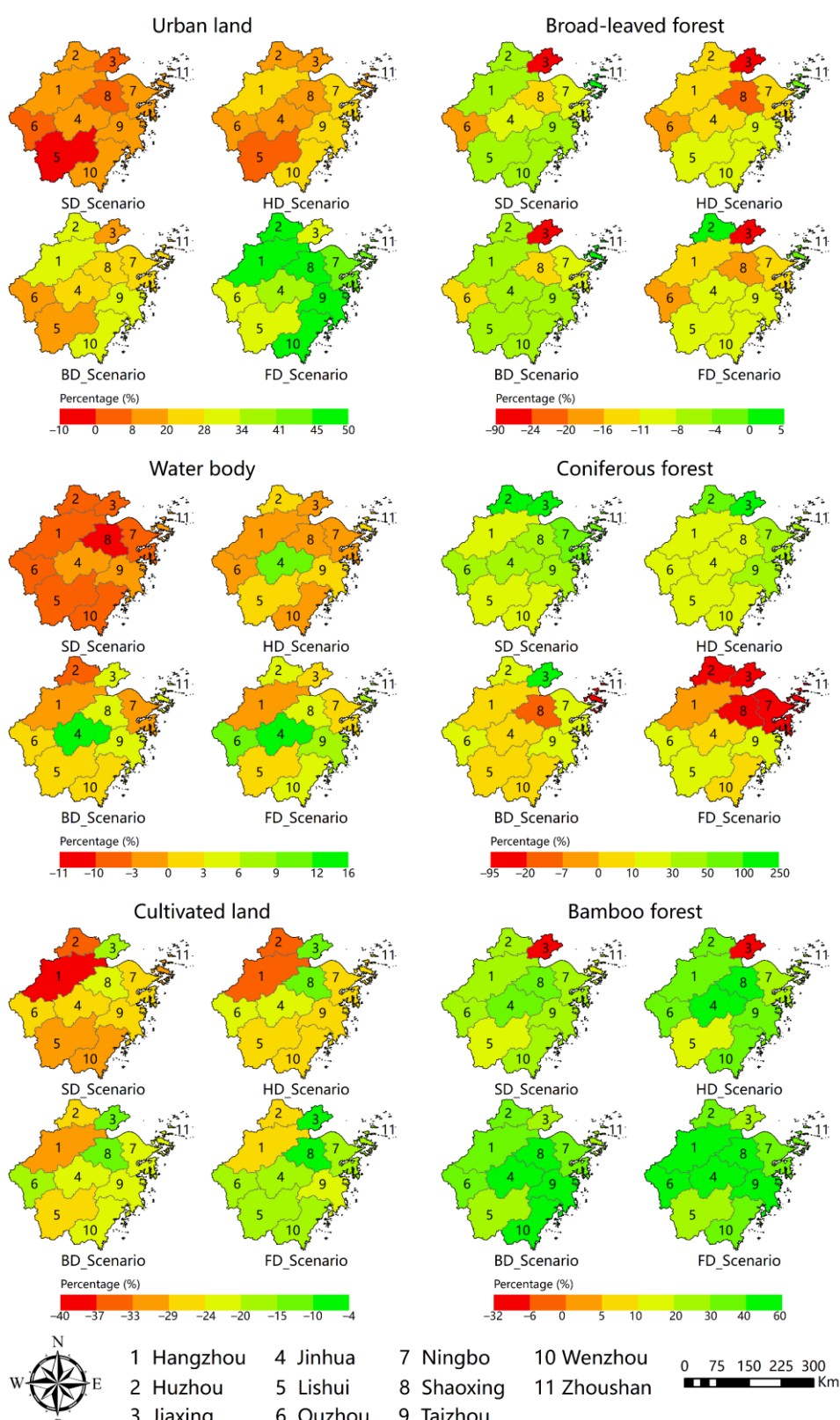

**Figure 14.** Land use/cover change amplitude at the municipal administrative level in Zhejiang Province from 2014 to 2084 under different future scenarios.

## 4. Discussion

### 4.1. Future Enhancements of the BCS Model

The reasons for the great simulation results based on the BCS coupled model are as follows: (1) the SD model considers complex factors such as social, economic, policy, and planning, and the uncertainty in predicting the quantitative area is reduced, thus achieving more accurate predictions than other quantitative models; (2) the CA model integrates a BPNN to extract extremely accurate land suitability considering more comprehensive driving data.

However, the model simulation still leaves some errors, mainly in the simulation of three forest types. For BLF, CF, and BF, the accuracy was 0.75, 0.69, and 0.73, respectively. Forest succession needs to take into account not only climate change and human disturbance but also its own growth conditions [55]. How to incorporate the spatial distribution of these factors for more accurate modeling and prediction is a challenge to overcome for our future research. Furthermore, parameters, such as land suitability and conversion costs, in the BCS coupled model are assumed to remain constant during the simulation, whereas these parameters will change over time in realistic situations [26]. Studies have already been conducted to predict the future spatial distribution of the population and economy on the basis of which future land suitability can be predicted [56]. Therefore, we need to invest more effort in improving the model based on the appropriate approach in future works.

### 4.2. Future Strategy for Land-Use Management

Different scenarios have different levels of ecological protection and different shares of investment in forestry; therefore, the forest coverage rate is higher in scenarios that focus on ecological protection. Consequently, the areas of BLF and CF are higher in the SD_Scenario and HD_Scenario than in the BD_Scenario and FD_Scenario after 70 years (Figure 11e,f). However, the reasons why the area of BF is higher in the FD_Scenario than in others are for the following two reasons: (1) The growth of BF is affected by the climate and has certain requirements for precipitation and temperature. The annual total precipitation required for BF growth is 1200–2500 mm, and the average daily temperature conducive to BF growth ranges from 15–25 °C [45]. Despite the increasing radiation intensity after the mid-21st century, in the FD_Scenario it results in increasing temperature and decreasing precipitation, and the temperature and precipitation in Zhejiang Province eventually ranged from 16.8–22.5 °C and 858.2–1693 mm, respectively (Figure 5a,b). Therefore, it can be recognized that Zhejiang Province will still be a natural environment suitable for bamboo forest growth in this future scenario. (2) In recent years, with the adjustment of the industrial structure of Zhejiang Province, the economic benefits of BF have become prominent. Moso bamboo and lei bamboo, with higher economic value, have become new growth points in the regional economy. The management intensity of BF has increased, and the area of BF has continued to increase. In the future FD_Scenario, the rapid development of the social economy will expand the management and investment of BF to a certain extent. The SD model feedback system takes this man-made activity into account; thus, the BF area obtained by simulation was larger than that simulated in other scenarios.

In the four scenarios, the simulated future LUCC evolutions were consistent with realistic changes. In the future, the proportion of each land-use type converted ranges from 0 to 23.27% (Figure 13), with the highest values being the conversion of BLF to CF in the HD_Scenario (23.27%) and the conversion of CL to UL in the FD_Scenario (19.74%). In the HD_Scenario, due to the impact of policies, in order to prevent the massive reduction of CF, the planned goals are achieved by converting part of the BLF into CF. In the FD_Scenario, the man-land contradiction will intensify, and CL will be mainly converted into UL, making it difficult to achieve the goal of protecting CL and posing a serious challenge to the ecological environment. LUCC is caused by complex and diverse factors, and the factors we can consider are limited. Therefore, it is likely that the future change will exceed our expectations; that is, the LUCC amplitudes simulated by the model may be smaller than the actual future change.

As seen in the future area change (Figure 11), the future spatial distribution (Figure 12), the LUCC conversion (Figure 13), and the change at the prefecture level (Figure 14), the SD_Scenario has less land-use change than the other scenarios, and this scenario is the more ecologically friendly one. Without considering the promotion of social and economic development, to maximize the restoration of forest ecology, and promote ecological protection, various indicators established in the SD_Scenario can be referred to for future land planning and forest resource management.

## 5. Conclusions

In summary, the aim of this study was to develop a BPNN_CA_SD (BCS) coupled model for future LUCC simulation and to analyze the LUCC of Zhejiang Province under different scenarios from 2014 to 2084. The BCS coupled model consisted of the BPNN_CA model and the SD model. The top-down SD model and the bottom-up BPNN_CA model were interactively integrated during the simulation. The simulation results in 2014 showed a great OA (0.8), Kappa (0.75), and relatively high FOM (>28%) value, indicating that the proposed model can simulate LUCC accurately. Under different scenarios, the future evolution of the LUCC simulated by the BCS model varied due to the different natural and human effects. By 2084, bamboo forests would increase by 37% under the FD_Scenario, while coniferous forests would decline by 25%. A comparison of the simulated subtropical forest area and spatial variation in the four scenarios revealed that the SD_Scenario was favorable to forest ecology. We also analyzed the future transfer area between land uses and the changes in each prefecture. These study results could provide an effective reference for decision makers regarding sustainable forest development and land-use planning under future climate conditions in Zhejiang Province.

**Author Contributions:** Conceptualization, H.D.; methodology, Z.H.; software, Z.H.; validation, Z.H., X.L. (Xuejian Li) and F.M.; formal analysis, Z.H., N.H. and W.F.; investigation, Z.H., Y.X. and X.L. (Xin Luo); data curation, Z.H.; writing—original draft preparation, Z.H.; writing—review and editing, H.D.; supervision, H.D. All authors have read and agreed to the published version of the manuscript.

**Funding:** This research was funded by the National Natural Science Foundation (No. 32171785, U1809208, 31901310), the State Key Laboratory of Subtropical Silviculture (No. ZY20180201), the Zhejiang Provincial Collaborative Innovation Center for Bamboo Resources and High-Efficiency Utilization (No. S2017011), and the Key Research and Development Program of Zhejiang Province (No. 2021C02005).

**Institutional Review Board Statement:** Not applicable.

**Informed Consent Statement:** Not applicable.

**Data Availability Statement:** Not applicable.

**Acknowledgments:** The authors gratefully acknowledge the support of various foundations. The authors are grateful to the editor and anonymous reviewers whose comments have contributed to improving the quality of this manuscript.

**Conflicts of Interest:** The authors declare no conflict of interest.

## Abbreviations

| Abbreviation | Description |
| --- | --- |
| LUCC | Land Use and Land Cover Change |
| UL | Urban Land |
| WB | Water Body |
| CL | Cultivated Land |
| BLF | Broad-Leaved Forest |
| CF | Coniferous Forest |
| BF | Bamboo Forest |
| CA | Cellular Automata |
| SD | System Dynamics |

| BPNN | Back Propagation Neural Network |
|------|-------------------------------|
| BPNN_CA | CA Model Integrated with the BPNN |
| BCS | BPNN_CA Model Integrated with the SD |
| CLUE-S | The Conversion of Land Use and Its Effects at the Small Regional Extent |
| OA | Overall Accuracy |
| Kappa | Kappa Coefficients |
| PA | Producer's Accuracy |
| ROC | Receiver Operating Characteristic |
| AUC | Area under ROC Curve |
| *FoM* | Figure of Merit |
| SD_Scenario | Slow Development Scenario |
| HD_Scenario | Harmonious Development Scenario |
| BD_Scenario | Base Development Scenario |
| FD_Scenario | Fast Development Scenario |
| RCP | Representative Concentration Pathway |
| CMIP5 | Coupled Model Intercomparison Project 5 |
| IPCC | Intergovernmental Panel on Climate Change |

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
