# Peer review of "Simulating Future LUCC by Coupling Climate Change and Human Effects Based on Multi-Phase Remote Sensing Data"

_remotesensing, doi:10.3390/rs14071698_

Round 1
Reviewer 1 Report
The article shows high merit in relation to a high amount of data handling with models and simulations.
The study and simulation appears very interesting for climate scientists and remote sensing researchers as well as modellers.
However, right in the abstract it would be nice to say for example line 19 that the approach tends to combine a machine learning / statistical model (period?), a spatio-temporal random model (cellular automata) and this system dynamics model (what is it? a land surface modelling? combined with a climate model GCM?). This could be made clearer in this abstract.
In the same way the kappa needs more information, i.e. based on which ground truth?
If the training was done with data from 1984 to 2014 (included or not?) the prediction just at 2014 is quite poor!
line 25 perhaps “a figure of merit” could be more appropriate without knowing what statistic or indice is used. (not “the”)
line 26 to predict land use or land cover?
line 30 these results are reflecting natural landscape (land cover) or managed land cover as part of forestry land uses?
Also, these categories seem to be the less well predicted (fig10C). The low score for FoM which seems to be a sort of total accuracy, needs to be discussed more (28% is quite low to get any confidence) I was wondering if a ground truth say at 2019 could be obtained to evaluate even more this confidence.
line 48 quantitative models aspects do not seem to consider climate models alternatives (land surface models) … or are they working at an appropriate time scale (e.g. simulation goals for up to 2084)? .. though on ines 90-95 this is mentioned with CMIP5 etc… without a clear link to the approach
The introduction finishes without knowing much more on the components (BPNN_CA and SD) coupled? Before, giving full details of these in the methodology part, it would be expected to get a feel of what they are and to be able already to be convinced of the originality and meaningfulness of the approach. For example, lines 198-199 would be better suited in the introduction.
The logic of figure 6 expressing the existing interactions beyond what is said in lines 198 and 199 would be required to fully grasp this workflow.
Something needs to be explained in relation to GDP and land use. I suppose not all GDP change comes from land use change?
Equation (1) is not very clear. What a state? a land use category I suppose? what “the overall probability” meaning here? Did you mean in fact that Pro is the vector of probabilities for all k? otherwise (1) is working only for a one k …? something is missing here?
Equation (2) is not fully described, … and/ or the numbering (1) and (2) is misleading here!
Do we have to understand that the land suitability os dependent only on RS data?
Line 305 really?
Line 309 , it is not clear where the R^2 comes from (which model?) The confusion matrix for 2014 in figure 9 would be interesting … or is it in figure 10C oh! and R2 is for the predicted areas of each land cover type! I wonder if this show more that the SD model is ok but not necessarily the the coupling or the bpmn-ca. IS the SD a forcing factor? Is this a scale effect, i.e. massive difference between categories and small changes for each land type?
Reviewer 2 Report
Type of manuscript: Article
Title: Simulating future LUCC by coupling climate change and human effects
based on multi-phase remote sensing data
Special Issue: Advanced Phenology, and Land Cover and Land Use Change Studies
The article is of great interest. The study shows great modeling potential for predict the land use, which can serve as an important decision-making reference for land use planning and forest sustainable development in Zhejiang Province. Abstract is well articulated. The goals set have been achieved.
However, the article has the potential for improvement and the Reviewer asks for individual points not only to give answers, but also to make the necessary adjustments. This primarily applies to the following comments.
A brief summary
- The authors used the retrospective series 1984-2014(5). But judging by Fig. 1c, for 8 years there has been a dynamic in the area of forests that cannot be ignored. Why hasn't newer data been brought in? The conclusions of this article are still relevant, the new forecast base will not change the qualitatively described results? The data in Table 3 "Annual growth rate settings from 2014 to 2084" is very confusing, since 2014-2021 is not a forecast period.
- Conclusions. L 465-474. This text literally repeats the text in Abstract.
- L 115. The authors first mention "sample plots in 2014". I would like to have answers with L 117: 1) what were the sample plots; 2) how many of them were there (is this the sum according to Table 2? (does the reader calculate for himself?), and why is there no statistical data on them (usual metrics for sampling)? 3) why was 2014? And not later than 2014-2020(1) ? (which is important, given the positive dynamics of forest areas in 2015-2020 (Fig. 1c). And also, in light of the fact that the basis - the retrospective basis - of the forecast strongly depends on the extreme dates! In addition, two indicators in Table 1 are linked to 2015.
- Table 1. Soil. Logical connection "forest - carbon sequestration - accumulation of OS in soils". The authors at L 44 write about the importance of this problem. Why is SOC (SOM) data not involved? Soil physics and hydrophysics is not all soil.
Specific Comments:
Abstract
L 19-20. The authors explain the abbreviation BPNN (back propagation neural network) later than it is given in L 19.
L 22. Appendix A. well, that shows all the abbreviations. But the Reader in the Abstract does not yet understand what is: HD_Scenario, BD_Sce- 22
nario, FD_Scenario).
L 24-27. It is clear that the results strongly depend on the quantity and quality of the input data. Therefore, the question to the authors is: Is the general scientific value of the results (point 1) that characterize the overall accuracy great?
Keywords. The Reviewer asks the authors to assess how two types of models - cellular automata (CA) model and system dynamics (SD) model - can be considered universal concepts and be a guide in scientific research?
L 34 (keywords) and 38. Compare: “land use and cover change (LUCC)” and “Land use and land cover change (LUCC)”
Fig. 12. The cardinal directions are indicated five times - this is already redundant.
References
Authors for a number of cases had to find abbreviated journal titles, as required by the RS requirements.
Authors should unify the Title of cited articles. That is, change Capital letters to lowercase for sources No. 1, 3, 6, 17, 19, 21, 31, 33, 34.
Description errors:
No 13. Science of The Total Environment
No 17. 2016, 2016
No 20. This source does not indicate the country? Proceedings of 539
the National Academy of Sciences
No 42. This source has no abbreviation? International Journal of Applied Earth Observation and Geoinformation

Reviewer 3 Report
I consider this work to be of very high quality because it used modern methods to estimate the future use of natural and human resources. The great importance arises from the possibility of planning activities for the management of natural and human resources in the future, especially due to climate change.
Round 2
Reviewer 1 Report
Even though the authors provided some answers to the series of questions and comments, I am left with the feeling there has not been enough chnages to fully answer them and improve the article.
The abstract and introduction din't particularly improve much and equation (1) still is not correct in relation to a vector of Pro ...(for example).
